# Aspalathin-Rich Green Rooibos Extract Lowers LDL-Cholesterol and Oxidative Status in High-Fat Diet-Induced Diabetic Vervet Monkeys

**DOI:** 10.3390/molecules24091713

**Published:** 2019-05-02

**Authors:** Patrick Orlando, Nireshni Chellan, Johan Louw, Luca Tiano, Ilenia Cirilli, Phiwayinkosi Dludla, Elizabeth Joubert, Christo J.F. Muller

**Affiliations:** 1Department of Life and Environmental Sciences, DiSVA-Biochemistry, Polytechnic University of Marche, 60131 Ancona, Italy; p.orlando@univpm.it; 2Biomedical Research and Innovation Platform (BRIP), South African Medical Research Council, Tygerberg 7505, South Africa; nireshni.chellan@mrc.ac.za (N.C.); johan.louw@mrc.ac.za (J.L.); pdludla@mrc.ac.za (P.D.); christo.muller@mrc.ac.za (C.J.F.M.); 3Division of Medical Physiology, Faculty of Health Sciences, Stellenbosch University, Tygerberg 7505, South Africa; 4Department of Biochemistry and Microbiology, University of Zululand, KwaDlangezwa 3886, South Africa; 5Department of Clinical Dental Sciences, Polytechnic University of Marche, 60131 Ancona, Italy; i.cirilli@pm.univpm.it; 6Plant Bioactives Group, Post-Harvest and Agro-Processing Technologies, Agricultural Research Council (ARC), Infruitec-Nietvoorbij, Private Bag X5026, Stellenbosch 7599, South Africa; joubertL@arc.agric.za; 7Department of Food Science, Stellenbosch University, Private Bag X1, Matieland 7602, South Africa

**Keywords:** type 2 diabetes, cardiovascular risk factors, antidiabetic activity, aspalathin-enriched green rooibos extract, coenzyme Q_10_

## Abstract

Type 2 diabetic patients possess a two to four-fold-increased risk for cardiovascular diseases (CVD). Hyperglycemia, oxidative stress associated with endothelial dysfunction and dyslipidemia are regarded as pro-atherogenic mechanisms of CVD. In this study, high-fat diet-induced diabetic and non-diabetic vervet monkeys were treated with 90 mg/kg of aspalathin-rich green rooibos extract (Afriplex GRT) for 28 days, followed by a 1-month wash-out period. Supplementation showed improvements in both the intravenous glucose tolerance test (IVGTT) glycemic area under curve (AUC) and total cholesterol (due to a decrease of the low-density lipoprotein [LDL]) values in diabetics, while non-diabetic monkeys benefited from an increase in high-density lipoprotein (HDL) levels. No variation of plasma coenzyme Q10 (CoQ_10_) were found, suggesting that the LDL-lowering effect of Afriplex GRT could be related to its ability to modulate the mevalonate pathway differently from statins. Concerning the plasma oxidative status, a decrease in percentage of oxidized CoQ_10_ and circulating oxidized LDL (ox-LDL) levels after supplementation was observed in diabetics. Finally, the direct correlation between the amount of oxidized LDL and total LDL concentration, and the inverse correlation between ox-LDL and plasma CoQ_10_ levels, detected in the diabetic monkeys highlighted the potential cardiovascular protective role of green rooibos extract. Taken together, these findings suggest that Afriplex GRT could counteract hyperglycemia, oxidative stress and dyslipidemia, thereby lowering fundamental cardiovascular risk factors associated with diabetes.

## 1. Introduction

Diabetes mellitus is a global pandemic afflicting 425 million adults worldwide, with trends suggesting the rate will continue to rise, reaching 642 million in 2040 [1]. Almost 90% of diabetic patients are affected by the lifestyle-related sub-type, type 2 diabetes (T2DM) [2], a progressive metabolic disease characterized by insulin resistance and eventual functional failure of pancreatic beta cells [3].

It is known that T2DM is associated with higher cardiovascular morbidity and mortality. In particular, type 2 diabetic patients have a two to four-fold increased risk of incident coronary heart disease, ischemic stroke and a 1.5 to 3.6-fold increase in mortality [4].

Hyperglycemia is a key cardiovascular risk factor for patients with type 2 diabetes [5]. Increased glucose flux through the polyol pathway, intracellular formation of advanced glycation end products (AGE) and increased expression of its receptors [5], as well as activation of protein kinase C (PKC) [6] and increased glucose flux through the hexosamine pathway [6,7,8] represent the five pro-atherogenic mechanisms of diabetes associated with hyperglycemia. Several lines of evidence indicate that these mechanisms are activated by mitochondrial reactive oxygen species (ROS) overproduction [9], produced by proton leakage at the mitochondrial electron transport chain, resulting in increased production of superoxide [10]. In turn, these mechanisms ultimately lead to further increases of free radical formation. Moreover, increased oxidative stress underlies endothelial dysfunction, caused by decreased bioavailability of nitric oxide (NO), a critical signaling molecule mediating vasoactive activity. In fact, high levels of ROS promotes NO oxidation, leading to the production of peroxynitrite, a highly reactive molecule responsible for extensive oxidative damage in the endothelium [11].

Besides hyperglycemia and oxidative stress, diabetic dyslipidemia—characterized by elevated plasma triglyceride concentrations, low-density lipoprotein (LDL) concentrations, and reduced high density lipoprotein (HDL) levels—represents one of the most common cardiovascular risk factors, with a prevalence of 72–85% in T2DM [12,13].

Further, macrophages have no affinity for non-oxidized LDL; even at high concentrations, non-oxidized LDL has little or no atherogenic-promoting properties. However, oxidized LDL contributes to the atherogenic processes in the arterial wall, driven by inflammation and the formation of lipid-laden foam cells that lead to plaque formation [14].

Accordingly, in order to prevent its oxidation, low plasma LDL levels should be maintained, particularly in clinical conditions characterized by elevated oxidative stress—such as in diabetes. Therefore, guidelines for primary and secondary prevention of cardiovascular disease in diabetes include hypocholesterolemic therapies for the reduction of plasma LDL [15]. Pharmacological targets endorsed by the American Diabetic Association refer to LDL plasma levels below 100 mg/dL in the primary prevention of individuals with diabetes [16], compared to 130 mg/dL in non-diabetic patients.

The Old World non-human primate species *Chlorocebus aethiops* (vervet monkey), endemic to Southern Africa, shares the same subfamily with the macaque (Cercopithecinae) and are phylogenetically close to humans [17]. Vervet monkeys are omnivorous and readily consume experimental diets, including high-fat diets, which has proven to be particularly useful for cardiovascular and metabolic disease research [18,19,20,21,22,23,24]. In particular, LDL concentrations correlate with the type and amount of fat in the diet [25]. A strong correlation was established within our vervet colony between dietary lipid intake, LDL cholesterol and atherogenic aortic lipid deposition [26]. An inherent susceptibility of this species to cholesterolemia and development of atherosclerotic lesions that correspond to the human classification types I–VII has also been confirmed [22,26]. In addition, these vervet monkeys are responsive to both dietary and pharmacological intervention strategies [18,21,22,23,24,27,28]. Although the normal plasma LDL:HDL ratio is lower in vervet monkeys [29,30] compared to humans [31], their LDL responses to Westernized human diets are similar and therefore relevant to this study [19].

*Aspalathus linearis*, also known as rooibos, is a shrub-like leguminous bush native to the Western and Northern Cape regions of South Africa. Commercially, rooibos is processed to produce unfermented (green) or fermented (red) rooibos. While fermentation results in oxidation of the plant polyphenols, leading to a decrease in the total antioxidant content, they remain preserved in green rooibos—particularly aspalathin, which represents the major bioactive compound [32].

In addition to its proven antioxidant properties, several studies demonstrated that rooibos tea has important anticancer [33], antihemolytic [34], antimutagenic [35,36] and anti-inflammatory [37] properties.

In the last years, many in vitro and ex vivo studies have focused on the antidiabetic effects of rooibos. In particular, Mazibuko et al. [38] confirmed the capacity of rooibos to reduce insulin resistance in C2C12 muscle cells, while other studies demonstrated a hypoglycemic effect of aspalathin in different murine type 2 diabetic models [39,40]. Finally, Kamakura et al. [41] used an aspalathin-rich green rooibos extract to promote glucose uptake in L6 myotubes and to counteract the increase in fasting blood glucose levels in type 2 diabetic model KK-Ay mice. The antidiabetic activity of rooibos is attributed to both the ability of aspalathin to increase GLUT 4 translocation to the plasma membrane via adenosine 5′ monophosphate-activated protein kinase (AMPK) and activation of Akt in skeletal muscle [38,41], and to reduce the gene expression of hepatic enzymes related to glucose production and lipogenesis [40].

Apart from a limited human study by Marnewick et al. [42], showing that consumption of rooibos tea (six cups/day) by participants at increased cardiovascular risk significantly decreased serum LDL and triglycerides and lowered plasma markers of lipid peroxidation, the effects of rooibos on other cardiovascular risk factors associated with diabetic pathology have been not studied.

This study aimed to evaluate the biological activity of standardized, pharmaceutical-grade, aspalathin-rich green rooibos extract (Afriplex, GRT) containing 12.8% aspalathin in improving the oxidative status and lipid profile of high-fat diet-fed diabetic and healthy non-human primates (*Chlorocebus pygerythrus*).

## 2. Results

### 2.1. Lipid Profile

Diabetic vervet monkeys showed significantly elevated plasma total cholesterol levels in comparison to normal monkeys (+164%; *p* < 0.01) at baseline, but also following treatment (14 days +101%; 28 days +123%; after wash-out +136%, *p* < 0.01) (Figure 1A). This is mainly due to different amounts of LDL (Figure 1B) between both experimental groups. Conversely, HDL levels are significantly different only at the start of the study (non-diabetics 1.59 ± 0.16 mmol/L, diabetics 2.44 ± 0.21 mmol/L; *p* = 0.010) (Figure 1C).

Two weeks of treatment with Afriplex GRT was sufficient to significantly decrease plasma LDL levels (baseline 6.64 ± 1.31 mmol/L; 14 days 5.27 ± 0.91 mmol/L, *p* = 0.015) (Figure 1B) and, consequently, total cholesterol (Figure 1A) (baseline 9.25 ± 1.11 mmol/L; 14 days 7.84 ± 0.73 mmol/L, *p* = 0.02) in diabetic monkeys. Moreover, after 4 weeks of treatment, LDL levels of diabetic monkeys remained unchanged (Figure 1B), while the total cholesterol significantly increased in comparison following 2 weeks treatment (Figure 1A) (+8%, *p* = 0.032). This is probably related to an increase, although not significant, of HDL levels in the same experimental group (Figure 1C) (+69%, *p* = 0.2).

In relation to non-diabetic vervet monkeys, total cholesterol and LDL levels remained unchanged at each experimental point (Figure 1A,B), while 4 weeks treatment promoted a highly significant increase of plasma HDL content in comparison with baseline (+118%, *p* = 0.012) and after 2 weeks of treatment (+90%, *p* = 0.008) (Figure 1C).

Summarizing, both diabetic and non-diabetic vervet monkeys showed a significant decrease in LDL:HDL ratio after supplementation with Afriplex GRT (Figure 1D), reducing the significant difference showed at baseline (non-diabetics 1.00 ± 0.12; diabetics 3.62 ± 0.97, *p* = 0.04). After the 4 weeks of wash-out, these values tended to revert back to the baseline level (Figure 1A–D).

In contrast, plasma triglyceride content remained unchanged in both studied populations (Figure 1E).

### 2.2. Glycemic Parameters

In order to evaluate insulin-response, fasting glucagon and glycemia levels, an intravenous glucose tolerance test (IVGTT) and a glucose-stimulated insulin secretion test (GSIST) were used.

As demonstrated in Figure 2A,B, at baseline diabetic animals showed significantly higher levels of glycaemia and insulin with respect to the non-diabetic monkeys (+63%, *p* = 0.03 and +125%, *p* = 0.04, respectively), while, in terms of fasting glucagon level, the difference recorded did not reach statistical significance (Figure 2C). Afriplex GRT did not affect the insulin or glucagon levels; despite this, after 2 weeks of treatment, Afriplex GRT had a noticeable hypoglycemic effect in the diabetic group, resulting in a significant decline of glycemia that persisted also after the wash-out period. More importantly, after 4 weeks of Afriplex GRT treatment, blood glycemia was reduced both in comparison to the baseline (from 487 ± 12 to 416 ± 4, *p* = 0.04) and the 2-weeks treatment (468 ± 7 to 416 ± 4, *p* = 0.03), also in the non-diabetic group.

### 2.3. Total Plasma Coenzyme Q10 (CoQ_10_) Level and Oxidative Status

CoQ_10_ plasma content was assessed in order to evaluate if the lipid-lowering effect of Afriplex GRT, highlighted in the diabetic group, also influenced CoQ_10_ synthesis via the mevalonate pathway. Interestingly, treatment did not produce any variation in plasma CoQ_10_ content in the diabetic monkeys, while non-diabetic monkeys showed a trend towards a decline in plasma CoQ_10_, which was restored after the wash-out period (Figure 3A).

Percentage of oxidized CoQ_10_ was also evaluated as a plasma oxidative marker in the monkeys. At baseline, the diabetic group showed a trend towards significantly higher levels of oxidized CoQ_10_ in comparison to the non-diabetics (Figure 3B) (non-diabetics 6.40% ± 0.01; diabetics 11.10% ± 0.02, *p* = 0.08). However, after Afriplex GRT treatment, the diabetic monkeys showed a significant decrease in the percentage of oxidized CoQ_10_; this effect also persisted after the wash-out period (Figure 3B).

### 2.4. Circulating Oxidized LDL

Circulating oxidized LDL (ox-LDL) was measured in plasma using an enzyme-linked immunosorbent assay (ELISA) kit. The diabetic monkeys showed significantly higher ox-LDL levels at each experimental time point, in comparison with the non-diabetic monkeys (Figure 4A). After 4 weeks of treatment, the diabetic monkeys showed a trend towards a decrease in this parameter (baseline 67.2 ± 14.8 U/L; 4-week treatment 53.8 ± 14.3 U/L), which returned to the baseline level after wash-out. However, a significant positive correlation was observed between ox-LDL and total LDL (R^2^ = 0.388, *p* = 0.017) (Figure 4B), while a significant inverse correlation was observed between oxidized LDL and lipoprotein CoQ_10_ content (mM CoQ/nM cholesterol) (R^2^ = 0.529, *p* = 0.002) (Figure 4C) in the diabetic monkey group.

## 3. Discussion

The present study focused on the effects of standardized pharmaceutical grade aspalathin-rich green rooibos extract (Afriplex GRT) in counteracting selected cardiovascular risk factors underlying diabetic pathology. Specifically, gluco-lipidic and oxidative indexes in high-fat diet-fed diabetic and non-diabetic vervet monkeys (*Chlorocebus pygerythrus*) were assessed prior to the onset of treatment (baseline), and after 2 and 4 weeks of treatment with 90 mg/kg of body weight of Afriplex GRT extract containing ca. 12.8% aspalathin. Moreover, analysis was repeated following 4 weeks of wash-out.

Clinically, the high-fat diet induced insulin resistance, glucose intolerance and, in susceptible monkeys, diabetic changes to the pancreatic islets; including reduced beta-cell mass and islet-associated amyloid deposits, exacerbating parameters for cardiovascular risk [43]. Although the monkeys on the high-fat diet (HFD) did not develop overt obesity in this colony, they displayed glucose intolerance with associated hyperinsulinemia and hyperglucagonemia, consistent with T2DM. In relation to glycemic parameters, a significant decrease of glycaemia was observed after just 2 weeks of treatment with Afriplex GRT and this effect persisted after 4 weeks of wash-out without any dietary interventions. It is important to note that there were no significant changes in the total calorie intake for the monkeys on their respective diets during the course of the study when compared to data recorded prior to the commencement of the study. This data, observed for the first time in non-human primates, confirms the hypoglycemic potential of green rooibos extract and its major phenolic compound, aspalathin, previously demonstrated in different murine type 2 diabetic models [39,40,41]. Additionally, the treatment effects were similar for males and females.

Interestingly however, insulin and glucagon levels remained unchanged in both the diabetic and non-diabetic animals. These results highlight that the potential antidiabetic role of Afriplex GRT could predominantly be related to an increase in insulin sensitivity, through promotion of glucose uptake, rather than its ability to stimulate the synthesis or secretion of the respective hormones. In this regard, Kamakura et al. [41] showed that the stimulatory effect of aspalathin on insulin secretion was less potent than the effect on glucose uptake in L6 myotubes; L6 myotubes responded to aspalathin concentrations 100 times lower than that which was effective in the beta cells.

In this study, besides the confirmatory effects on glucose usage in diabetic models, and in line with other reports in the literature, we also showed a marked improvement on the plasma lipid profile. In both the diabetic and non-diabetic vervet monkeys, the LDL:HDL ratio, a cardiovascular risk indicator with greater predictive value than isolated indexes used independently, were improved [44]. Vervet monkeys on a high-fat diet have an inherent susceptibility to developing LDL-cholesterolemia and atherosclerosis comparable to that found in humans [19,25,26].

Specifically, treated non-diabetic monkeys benefited from an increase of plasma HDL levels, whose anti-atherogenic properties are widely demonstrated [45], while diabetic animals mainly benefited from a decrease in LDL plasma levels that reached significant levels in only 2 weeks of treatment. In a previous study, treating high-fat fed monkeys with etofibrate for 20 months lowered LDL and demonstrated an anti-atherogenic effect [26]. In addition to lowering LDL, etofibrate has been demonstrated to protect LDL against oxidation in human subjects, and thereby lower cardiovascular risk in diabetic patients [46]. In a human study, Marnewick et al. demonstrated a decrease of 15% in circulating LDL and an increase of 33% in HDL as well as redox status following consumption of six cups of rooibos tea daily for six weeks [42]. These findings validate the relevance of this diet-induced vervet monkey model in predicting treatment efficacy of human metabolic disease.

This remarkable ability of rooibos flavonoids to improve lipid profiles could play an important role in cardiovascular protection of diabetic patients. In fact, it is known that the main pharmacological approaches used in the primary and secondary prevention of cardiovascular diseases associated with diabetic dyslipidemia aim to lower circulating LDL levels. In particular, 3-hydroxy-methylglutaryl coenzyme A (HMG-CoA) reductases inhibitors, also known as statins, are the most common and consolidated therapy used for this purpose. However, even though their efficiency to reduce cardiovascular morbidity and mortality in diabetes has been widely demonstrated [47], statins can cause adverse side effects. Primarily, they may promote hyperglycemia by increasing the calcium concentration in pancreatic β islet cells, leading to a decrease in insulin release, or by decreasing GLUT 4-mediated peripheral glucose uptake [48]. Secondarily, multiple studies have shown that statins can decrease serum coenzyme Q_10_ levels [49], affecting an early step in the mevalonate pathway. In fact, mevalonate is a common precursor for cholesterol and CoQ_10_ synthesis. CoQ_10_, also known as ubiquinone, is an endogenous quinone with a key role in mitochondrial bioenergetics; in its reduced form (ubiquinol), it represents one of the most important lipophilic antioxidants [50]. Observational studies have reported that plasma CoQ_10_ concentration is an independent predictor of mortality in patients with congestive heart failure [51], whereby its decline associated with statin therapy may be related not only to muscle disorders but also to increased risk of cardiovascular diseases. These indications support the hypothesis that statins, while efficiently minimizing some CVD related risk factors, may be detrimental to cardiovascular health in diabetic patients by promoting hyperglycemia and decreasing CoQ_10_ biosynthesis.

In our study, while the treatment with 90 mg/day/kg of body weight of pharmaceutical grade aspalathin-enriched green rooibos extract (Afriplex GRT) was effective in lowering serum LDL levels in diabetic animals, no gross side effects described for statin treatment was observed. On the contrary, no variation in terms of plasma CoQ_10_ normalized for cholesterol was observed during all experimental phases in diabetic monkeys. This suggests that the LDL-lowering effect of aspalathin could be related to its ability to modulate the mevalonate pathway differently from statins, or to affect other biochemical mechanisms not affecting CoQ_10_ biosynthesis, therefore preserving its cardioprotective role.

In fact, several studies showed that some flavonoids directly affect cholesterol metabolism at different steps: Curcumin is able to increase the excretion of bile acid by upregulating the expression of cholesterol 7α-hydroxylase [52]; tocotrienols inhibit HMG-CoA reductase [53,54] similarly to statins; and flavonoids of green tea upregulate the LDL receptor [55,56]. Beltrán-Debón et al. [57] demonstrated a hypolipidemic effect of rooibos in LDLr-/-mice fed a high-fat Western-type diet, but this effect was stringently dependent on diet type. In these high-fat fed animals, rooibos extract significantly lowered serum cholesterol, triglyceride and free fatty acid concentrations. However, mechanisms underlying LDL-lowering effect of Afriplex GRT are not reported in literature.

Finally, in order to evaluate the antioxidant role of Afriplex GRT treatment, the plasma oxidative status of CoQ_10_ was analyzed. In fact, several studies demonstrated that the reduced form of CoQ_10_, a key and ubiquitous lipophilic antioxidant, could represent a sensitive marker of oxidative stress in vivo [58,59,60,61]. At baseline, diabetic animals showed a higher—although not significant—percentage of oxidized CoQ_10_ in comparison to non-diabetic monkeys. Only 2 weeks of treatment with Afriplex GRT was sufficient to abolish this difference, significantly decreasing the percentage of oxidized CoQ_10_. Remarkably, this antioxidant effect persisted after 4 weeks of wash-out. In line with the improvement of plasma oxidative status, a slight, although not significant, decrease of circulating ox-LDL was observed following treatment in the diabetic group, although after 4 weeks of wash-out these were restored to baseline values. The significant correlation between the amount of circulating ox-LDL and total LDL concentration observed in the diabetic population confirms, in a pro-oxidant condition characterizing diabetic pathology, the high susceptibility of lipoproteins to oxidation and the relevance of hypocholesterolemic therapy. However, the significant reverse correlation between circulating ox-LDL and plasma CoQ_10_ levels detected in the same population suggests that CoQ_10_ could play an important role in protecting LDL from oxidation, as is widely demonstrated in literature [62,63,64,65].

## 4. Material and Methods

### 4.1. Vervet Monkeys: Ethics

In-house bred vervet monkeys (*Chlorocebus pygerythrus*) were cared for and managed according to the documented standard operating procedures of the Primate Unit and Animal Centre (PUDAC) of the South African Medical Council, and according to the SAMRC Guidelines for the Use of Animals in Research and Training, the National Code for Animal Use in Research, Education, Diagnosis and Testing of Drugs and Related Substances in South Africa, and the Veterinary and Para-Veterinary Professions Act of 1997. Ethical approval numbers 12/03 and 01/16 were obtained from the Ethics Committee for Research on Animals (ECRA) of the South African Medical Research Council.

### 4.2. Aspalathin-Rich Green Rooibos Extract

In this study, a pharmaceutical certified grade, aspalathin-rich, unfermented rooibos (Afriplex GRT™) extract produced by Afriplex Pharmaceuticals PTY (LTD) (Paarl, South Africa), product code: CPE—03287, Batch Number: 730330, manufactured date: September 2015 and expiry date: August 2017, was used. The extract was previously chemically characterized for its phenolic content and was shown to contain approximately 12% aspalathin [66]. The selected dose of 90 mg/kg BW was calculated from a dose range (25–300 mg/kg) shown to be effective in rats and extrapolated to monkeys using the method described by Reagan-Shaw et al. [67].

### 4.3. Experimental Design

Eight vervet monkeys (*Chlorocebus pygerythrus*) (4 male/4 female, age 18 ± 2, BMI 5.0 ± 0.3) on a high-fat, diabetogenic diet (1527 kJ per monkey per day with 14% energy from protein, 43% from fat derived from animal and plant sources (99 mg/day cholesterol with a polyunsaturated/saturated [P/S] fat ratio of 0.3 and 43% from carbohydrates) for at least 5 years prior to selection and six normal controls (4 male/2 female, age 16 ± 1, BMI 4.3 ± 0.3) fed a maize based control diet (1378 kJ per monkey per day with 15% energy from protein, 5% fat as energy (19.7 mg/day cholesterol with a P/S fat ratio of 3.40) and 80% from carbohydrate) were selected and maintained at the Primate Unit (PUDAC) of SAMRC. They had access to water ad lib via an automatic watering device. The respective diets were continued for the duration of the study, including the wash-out period.

Both diabetic and non-diabetic monkeys were treated for four weeks with Afriplex GRT, fed orally to the monkeys at a dose of 90 mg/kg BW with a 30 g bolus of maize three times daily, followed by 1-month wash-out. Their demographic data are presented in Table 1. Blood was collected by femoral venipuncture under ketamine anesthesia at 10 mg/kg body weight intramuscular injection after overnight fasting of at least 14 h. Blood was collected at baseline (before treatment), and during treatment at 2- and 4-weeks, and following a 4 weeks wash-out period. Analysis included the plasma lipid profile, fasting plasma glucose and intravenous glucose tolerance test (IVGTT), glucose stimulated insulin secretion test (GSIST), fasting glucagon levels, plasma CoQ_10_ levels, CoQ_10_ oxidative status and oxidized-LDL levels.

### 4.4. Lipid Profile

Lipid profiles, in terms of total cholesterol, LDL, HDL and triglyceride levels, were determined in plasma by Pathcare Laboratories (N1 City, Cape Town, South Africa) using a Beckman AU5800 analyzer for lipid analyses. Specifically, an enzymatic method (using cholesterol esterase) was used to determine cholesterol concentrations, a coupled enzymatic reaction (using ATP as an agent) was used for triglyceride determination, an enzyme chromogen reaction was used for HDL determination, and a cholesterol esterase/cholesterol oxidize method was used for LDL determination. Results are expressed as mmol/L and as an LDL:HDL ratio.

### 4.5. Total Plasma CoQ_10_ Levels and Oxidative Status

CoQ_10_ content and its oxidized form in plasma were assayed by high performance liquid chromatography (HPLC) with electro-chemical detector (ECD) by Shiseido Co. Ltd. (Tokyo, Japan) as described by Orlando et al. [68]. Plasma levels of CoQ_10_ were normalized for total cholesterol, and expressed as nmol of CoQ_10_/mmol of cholesterol, representing the CoQ_10_ in plasma mainly associated with LDL. The oxidative status of CoQ_10_ was reported as percentage of oxidized CoQ_10_/total CoQ_10_.

### 4.6. Circulating Oxidized-LDL

Circulating oxidized-LDL (ox-LDL) was measured in plasma by a sandwich ELISA using mAb-4E6 as a specific monoclonal antibody. The assay was conducted according to the custom protocol (Oxidized LDL ELISA kit, Mercodia, Sweden), and the results are expressed as U/L.

### 4.7. Glycemic Responses

After drawing the baseline blood samples, 50% dextrose (750 mg/kg BW) was infused via the saphenous vein. Blood samples for glucose were collected into sodium fluoride/potassium oxalate tubes at 0, 5, 10, 15, 20, 40 and 60 min. To assess the glucose stimulated (GS)-insulin response, blood samples were collected at 0, 5, 10, 15, 30 and 60 min into serum separating tubes (SST), and centrifuged at 2000× *g* for 10 min in a refrigerated centrifuge after clotting for 30 min. Blood for assessing fasting glucagon were collected into ethylenediaminetetraacetic acid (EDTA) tubes. Results are expressed as area under curve (AUC) for IVGTT and GSIST, and as pg/mL for glucagon.

### 4.8. Statistical Analysis

For data analysis, mean value, standard deviation and standard error of means (SEM) were calculated. All values were presented as means ± SEM. The normal distribution of the data was verified using the Shapiro–Wilk test on the pooled data for each parameter. Homoscedasticity was verified using the Bartlett test, thereby comparing the variance within each group. Both assumptions were verified and therefore a parametric test statistical analysis of the data was performed. The significance of differences between mean values obtained from both experimental groups and before and following Afriplex GRT treatment were evaluated using one way ANOVA with Dunnett post hoc test if significant. A *p*-value < 0.05 was considered statistically significant, and *p* < 0.01 was considered highly significant. Pearson’s correlation coefficient was used to evaluate the correlations between ox-LDL and both LDL and CoQ_10_/Chol plasma levels. Area under the curve values were calculated in GraphPad Prism version 6 using the trapezoid rule.

## 5. Conclusions

In conclusion, the present study demonstrated that treatment of diabetic vervet monkeys with 90 mg/day/kg of body weight of pharmaceutical-grade aspalathin-enriched green rooibos extract containing 12.8% aspalathin could counteract hyperglycemia, oxidative stress and dyslipidemia, which represent the main cardiovascular risk factors in diabetic pathology. In addition, the cardioprotective role of Afriplex GRT is emphasized by its ability to lower serum LDL levels, preserving CoQ_10_ biosynthesis and therefore maintaining its important cardiovascular role.

In light of these results, further studies are needed to investigate the mechanism by which aspalathin lowers serum lipids, and to test its efficacy in association with statins for the purposes of achieving a pharmacological target of plasma LDL at lower doses, thereby minimizing the potential toxic side effects of these widely used drugs.

## Figures and Tables

**Figure 1 molecules-24-01713-f001:**
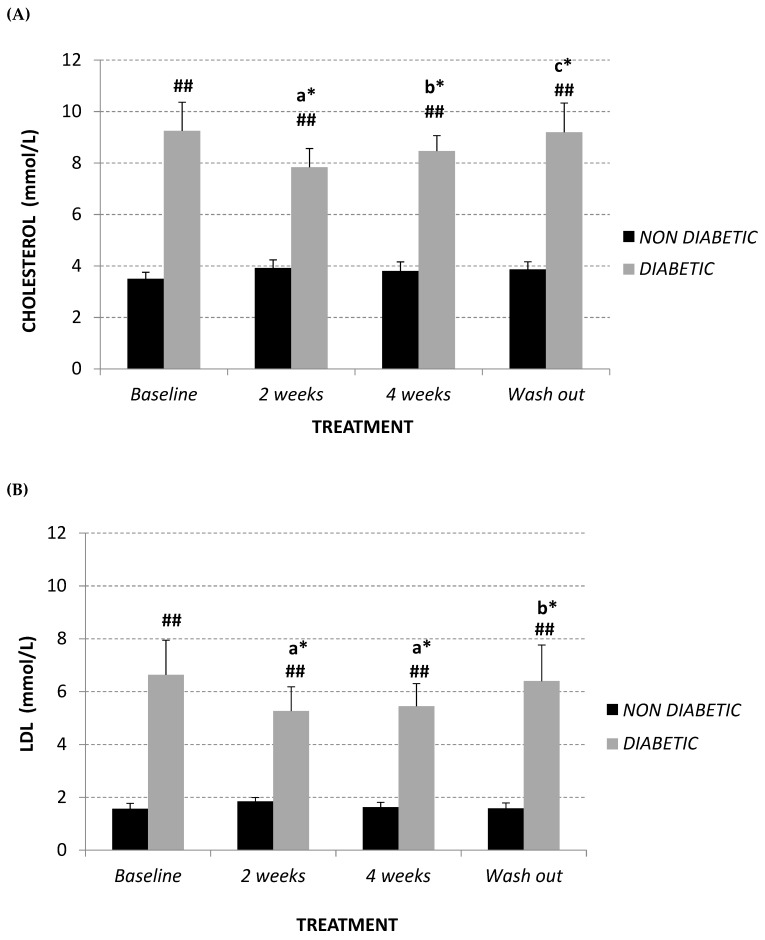
Plasma total cholesterol (**A**), LDL (**B**), HDL (**C**) (mmol/L), LDL/HDL ratio (**D**) and triglyceride (mmol/L) (**E**) at baseline, after 2 and 4 weeks of Afriplex GRT treatment, and following 4 weeks wash-out in non-diabetic (black) and diabetic (grey) vervet monkeys. * *p* < 0.05, ** *p* < 0.01 comparing different time points within each experimental group (a = baseline, b = 2 weeks-treatment, c = 4 weeks-treatment); ^#^
*p* < 0.05 and ^##^
*p* < 0.01 comparing both population groups for the same experimental point. a = baseline, b = 2 weeks-treatment, c = 4 weeks-treatment.

**Figure 2 molecules-24-01713-f002:**
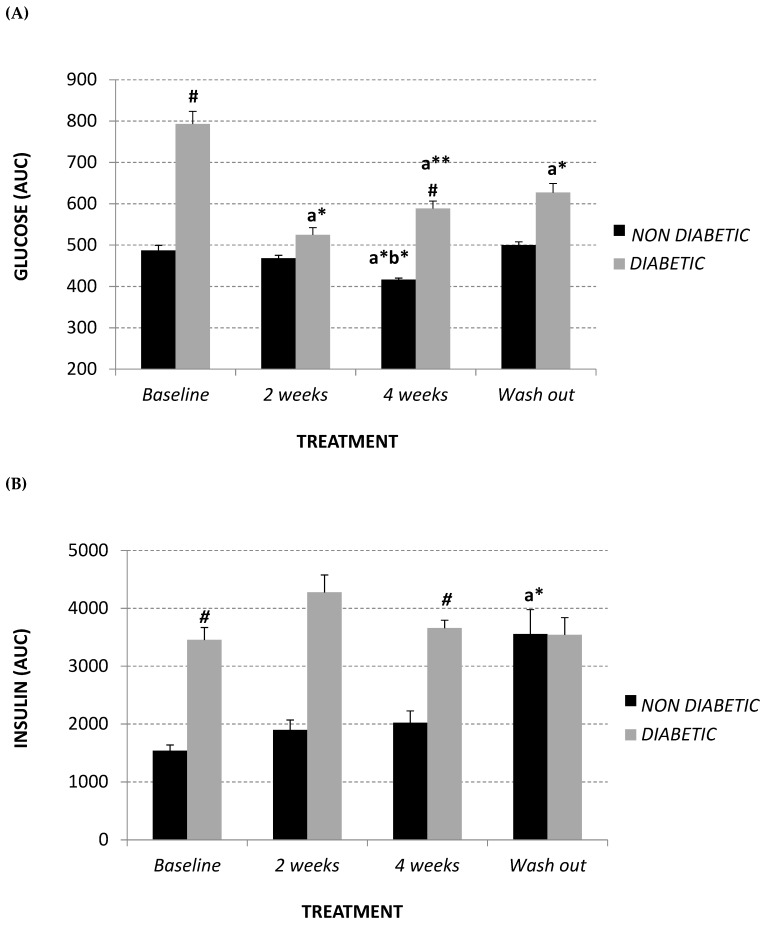
Intravenous glucose tolerance test (area under curve: AUC) (**A**), glucose stimulated insulin secretion test (AUC) (**B**) and fasting glucagon (pg/mL) (**C**) at baseline, after 2 and 4 weeks of Afriplex GRT treatment, and following 4 weeks wash-out in non-diabetic (black) and diabetic (grey) vervet monkeys. * *p* < 0.05, ** *p* < 0.01 comparing different experimental time points for each group; ^#^
*p* < 0.05 comparing both population groups for the same experimental time point. a = baseline, b = 2 weeks-treatment.

**Figure 3 molecules-24-01713-f003:**
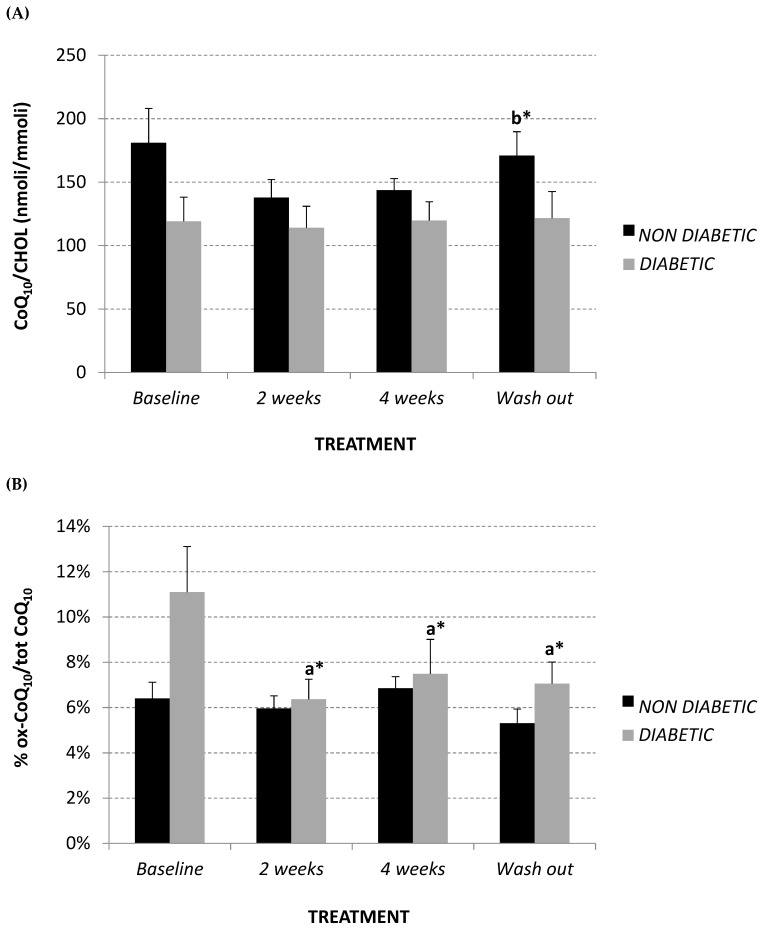
Plasma CoQ_10_/Cholesterol levels (nmol/mmol) (**A**), and % oxidized CoQ_10_/total (**B**) at baseline, after 2 and 4 weeks of Afriplex GRT treatment, and following 4 weeks wash-out in non-diabetic (black) and diabetic (grey) vervet monkeys. * *p* < 0.05 comparing different experimental points for each group. a = baseline, b = 2 weeks-treatment.

**Figure 4 molecules-24-01713-f004:**
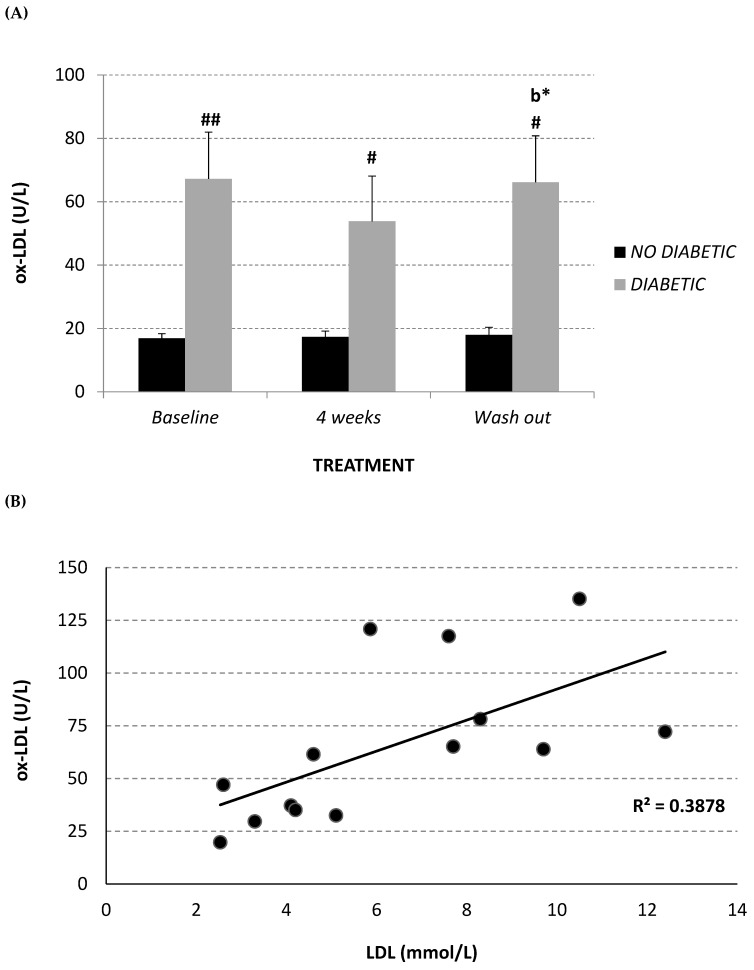
Plasma oxidized LDL level (U/L) (**A**) at baseline, after 2 and 4 weeks of Afriplex GRT treatment, and following 1 month-wash-out in non-diabetic (black) and diabetic (grey) vervet monkeys. Correlation between plasma oxidized LDL level and both total LDL (**B**) and CoQ_10_/Chol level (**C**) in diabetic monkeys in baseline, after 4 weeks of supplementation and wash-out period. * *p* < 0.05 comparing different experimental points for each group; ^#^
*p* < 0.05 ^##^
*p* < 0.01 comparing both groups for the same experimental time point. b = 4 weeks-treatment.

**Table 1 molecules-24-01713-t001:** Demographic data of vervet monkeys.

Monkey Groups		Gender	Date of Birth	Age at Baseline (years)	Body Weight During Study (kg)			
					*Baseline*	*2 Weeks*	*4 Weeks*	*Wash-Out*
**Non-diabetic**	**M268**	Male	25-set-00	14	4.7	4.6	4.5	4.8
	**M1077**	Male	Wild caught	Mature adult	4.9	4.9	4.8	5.0
	**M248**	Male	18-set-99	16	4.4	4.3	4.3	4.4
	**M1068**	Male	Wild caught	Mature adult	4.8	4.8	4.8	4.9
	**M205**	Female	13-mag-98	17	3.3	3.3	3.3	3.3
	**M234**	Female	14-apr-99	16	3.7	3.8	3.8	3.8
**Diabetic**	**M49**	Female	27-ago-90	25	3.6	3.6	3.6	3.5
	**M343**	Male	21-mar-03	12	5.9	6.0	5.9	5.7
	**M281**	Male	31-gen-01	14	5.5	5.7	6.1	5.7
	**M1083**	Male	01-giu-99	16	6.1	6.3	6.5	6.2
	**M403**	Female	Wild caught	Mature adult	5.2	5.2	5.3	5.3
	**M39**	Female	19-ott-89	25	4.6	4.8	4.9	4.8
	**M238**	Male	19-mag-99	16	5.1	5.0	5.1	5.3
	**M136**	Female	01-feb-95	20	4.0	3.9	3.7	3.8

## Data Availability

All data used to support the findings of this study are available from the corresponding author upon request.

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
