# Peer review of "Aspalathin-Rich Green Rooibos Extract Lowers LDL-Cholesterol and Oxidative Status in High-Fat Diet-Induced Diabetic Vervet Monkeys"

_molecules, 2019, doi:10.3390/molecules24091713_

Round 1
Reviewer 1 Report
This is an interesting manuscript which orlando et al report that 90 mg/kg of aspalathin-rich green rooibos extract (Afriplex GRT) can decrease plasma LDL, and cholesterol, and improved IVGTT glycemic AUC, decreased the percentage of oxidized CoQ10, and circulating ox-LDL levels in diabetes. The statistical assay is not clear for each fig. Fig 2B plasma Insulin AUC does have significant between 2 weeks to baseline. The discussion is too long, not point.
The manuscript looks like it had been cut and pasted from
different sources, since the paper has multiple fonts and/or font size (line 369-line
379)
The paper uses a large number of acronyms and abbreviations. Some of these are
described but many are not and/or abbreviated incorrectly. Undefined
abbreviations include LDL. HDL IVGTT, GSIST.
Author Response
REVIEWER 1
This is an interesting manuscript which orlando et al report that 90 mg/kg of aspalathin-rich green rooibos extract (Afriplex GRT) can decrease plasma LDL, and cholesterol, and improved IVGTT glycemic AUC, decreased the percentage of oxidized CoQ10, and circulating ox-LDL levels in diabetes.
The statistical assay is not clear for each fig.
The authors explained the statistical analysis of data in the section 4.8 of the manuscript. However, in order to clear the graphic representation, they had improved the legend of some figures in the manuscript.
Fig 2B plasma Insulin AUC does have significant between 2 weeks to baseline.
Although diabetic population showed an increase of insulin levels after two weeks of aspalathin-rich green rooibos extract supplementation, this difference is not statistically significant (baseline=3454±214; 2-weeks=4276±301; p=0.18).
The discussion is too long, not point.
Accordingly to the reviewer comment, the discussion of manuscript has been slightly shortened removing from line 288 to 293.
The manuscript looks like it had been cut and pasted from different sources, since the paper has multiple fonts and/or font size (line 369-line 379)
Accordingly to the reviewer comment, the authors have standardized the format of the manuscript.
The paper uses a large number of acronyms and abbreviations. Some of these are described but many are not and/or abbreviated incorrectly. Undefined abbreviations include LDL. HDL IVGTT, GSIST.
Accordingly to the reviewer comment, the authors have defined LDL and HDL in line 65. In addition, previously IVGTT and GSITS were defined in lines 351, but these abbreviations were used for the first time in line 161 of the manuscript. In this sense the authors have corrected the error.
Reviewer 2 Report
In this study, Orlando et al. report their findings on their study investigating the effects of Aspalathin, a bioactive compound found in green rooibos extract, on the lipid and glucose profiles of diabetic and non-diabetic Vervet monkeys. The authors show interesting results on changes in lipid profiles (especially an increase in HDL levels in both groups) and a decrease in blood glucose that is independent of insulin and glucagon levels. The abstract, introduction, and discussion are all very clear, following a logical flow, to the point and very well-written. On the other hand, results section needs a little more details, for improving clarity for the reader. I think the study in general is of interest to the readership of Molecules and I have the following comments and suggestions to help improve the manuscript, especially on the clarity of experimental design and discussion on potential confounding factors.
Introduction
The authors indicate that Vervet Monkeys are closer to humans in their lipid profiles and atherosclerosis development. Do they have also similar ratios of HDL/LDL? For instance, this is a major issue in studies using mice as model organism and the authors should emphasize the superiority of their experimental animal model by giving such numbers.
Methods, Results & Discussion
The authors indicate the monkeys were kept on the corresponding diets (control and high-fat diet (HFD)) for 5 years prior to the experiments. Were they then kept on the same diet for the duration of the experiment also? I assume this is the case, however, this information missing in the methods section (could be added somewhere around line 336), as the change in diet could explain some of the differences seen. In fact, even if the monkeys were kept on the same diet as before, the administration of Afriplex GRT 3 times a day with a bolus of 30 g of maize might have changed the eating habits of the monkeys slightly (reduced appetite for regular food). The authors should discuss this possibility in the discussion.
Did the monkeys on HFD develop also obesity? What was the criteria for determining that the monkeys were diabetic? These values and descriptions are important to add as not all readers (including myself) are experts on Vervet Monkey physiology.
The authors used male and female monkeys in the study, was there a difference in the responses of different genders?
What was the source of fat in the high-fat diet? Please indicate in methods.
I think what would be most helpful is the data to be presented also in a table format, including demographic data of the animals (age, weight etc) and presenting mean values plus/minus the standard deviation in one place. This would be greatly helpful to compare the effects to each other in one place.
Figures 1A-1D. It seems that the total cholesterol is decreased maximum about 10% in the non-diabetic monkeys over the course of treatment and it bounces back to the previous levels in the wash out period. How is this effect as compared to statins? Is 10% decrease in total cholesterol biologically significant? More interestingly, there is a doubling of HDL cholesterol at 4 weeks. What do the authors think the mechanism for this could be? Perhaps more than the lowering of total cholesterol, an increase in HDL might have a greater effect on the reduction of atherosclerosis via engaging reverse cholesterol transport mechanisms, therefore it would be important to understand how this pathway is triggered.
Figure 2A, it seems there is a great decrease of blood glucose in the diabetic monkeys, without major changes in insulin production. As for glucagon, the authors say there was no difference at the baseline. I understand that the difference might not be statistically significant, but the glucagon levels in diabetic monkeys is 2X of that of the non-diabetic group. I would take a more cautious approach and indicate that there was a difference which didn’t reach statistical significance. I think these results warrant future studies investigating the glucose uptake into muscle during Afriplex GRT intake.
Minor comment
Throughout the text the experimental groups are referred to as “Diabetic” and “Non-diabetic”. This sounds appropriate, therefore I suggest the legends shown in the graphs should be changed from “No diabetic” to “Non-diabetic”
Recommended
Author Response
REVIEWER 2
In this study, Orlando et al. report their findings on their study investigating the effects of Aspalathin, a bioactive compound found in green rooibos extract, on the lipid and glucose profiles of diabetic and non-diabetic Vervet monkeys. The authors show interesting results on changes in lipid profiles (especially an increase in HDL levels in both groups) and a decrease in blood glucose that is independent of insulin and glucagon levels. The abstract, introduction, and discussion are all very clear, following a logical flow, to the point and very well-written. On the other hand, results section needs a little more details, for improving clarity for the reader. I think the study in general is of interest to the readership of Molecules and I have the following comments and suggestions to help improve the manuscript, especially on the clarity of experimental design and discussion on potential confounding factors.
INTRODUCTION
The authors indicate that Vervet Monkeys are closer to humans in their lipid profiles and atherosclerosis development. Do they have also similar ratios of HDL/LDL? For instance, this is a major issue in studies using mice as model organism and the authors should emphasize the superiority of their experimental animal model by giving such numbers.
HDL/LDL ratio in Vervet monkeys is slightly lower compared to humans as confirmed by Kavanagh et al. (Characterization and heritability of obesity and associated risk factors in vervet monkeys. Kavanagh K, Fairbanks LA, Bailey JN, Jorgensen MJ, Wilson M, Zhang L, Rudel LL, Wagner JD. Obes Silver Spring Md. 2007 Jul;15(7):1666–74) and van Jaarsveld et al. (van Jaarsveld PJ, Benadé AJS. Effect of palm olein oil in a moderate-fat diet on low-density lipoprotein composition in non-human primates. Asia Pac J Clin Nutr. 2002;11 Suppl 7:S416–23). In particular, in the present study these animals showed a value of 1.00 ± 0.12 and 3.62 ± 0.97 in non-diabetic and diabetic group, respectively. In humans HDL/LDL ratio is about 3 in normal condition and 5 in dyslipidemic patients (Health Care Spending in the United States and Other High-Income Countries. Papanicolas I., Woskie R.L., Jha A.K. JAMA. 2018;319(10):1024-1039). However, specifically their LDL responses to human westernized diets are similar and therefore relevant to this study, as described by Weight et al. (Weight MJ1, Benade AJ, Lombard CJ, Fincham JE, Marais M, Dando B, Seier JV, Kritchevsky D. Low density lipoprotein kinetics in African Green monkeys showing variable cholesterolaemic responses to diets realistic for westernised people. Atherosclerosis. 1988 Sep;73(1):1-11). This concept has been added in the “Introduction” section of the manuscript and the references has been updated.
METHODS, RESULTS & DISCUSSION
The authors indicate the monkeys were kept on the corresponding diets (control and high-fat diet (HFD)) for 5 years prior to the experiments. Were they then kept on the same diet for the duration of the experiment also? I assume this is the case, however, this information missing in the methods section (could be added somewhere around line 336), as the change in diet could explain some of the differences seen. In fact, even if the monkeys were kept on the same diet as before, the administration of Afriplex GRT 3 times a day with a bolus of 30 g of maize might have changed the eating habits of the monkeys slightly (reduced appetite for regular food). The authors should discuss this possibility in the discussion.
Accordingly to the reviewer comment, additional information requested has been entered in lines 355-356 of “Experimental design” and in lines 240-243 of “Discussion” section.
Did the monkeys on HFD develop also obesity? What was the criteria for determining that the monkeys were diabetic? These values and descriptions are important to add as not all readers (including myself) are experts on Vervet Monkey physiology.
Accordingly to the reviewer comment, additional information requested has been entered in lines 236-238 of “Discussion”.
The authors used male and female monkeys in the study, was there a difference in the responses of different genders?
No, the treatment effects were similar for the males and females. This indication has been added in line 246.
What was the source of fat in the high-fat diet? Please indicate in methods.
Accordingly to the reviewer comment, additional information requested has been entered in lines 349-350 of “Experimental design” section.
I think what would be most helpful is the data to be presented also in a table format, including demographic data of the animals (age, weight etc) and presenting mean values plus/minus the standard deviation in one place. This would be greatly helpful to compare the effects to each other in one place.
Accordingly to reviewer comment, the authors have added the table including demographic data of monkeys in the “Experimental design” section.
Figures 1A-1D. It seems that the total cholesterol is decreased maximum about 10% in the non-diabetic monkeys over the course of treatment and it bounces back to the previous levels in the wash out period. How is this effect as compared to statins? Is 10% decrease in total cholesterol biologically significant? More interestingly, there is a doubling of HDL cholesterol at 4 weeks. What do the authors think the mechanism for this could be? Perhaps more than the lowering of total cholesterol, an increase in HDL might have a greater effect on the reduction of atherosclerosis via engaging reverse cholesterol transport mechanisms, therefore it would be important to understand how this pathway is triggered.
Maybe the reviewer refers to diabetic monkeys. In this population a significant decrease of total cholesterol mainly due to a strong decline of LDL component was occurred. In this regard, the ability of statin treatment to decrease both indexes is closely related to dose, drug type used and lipid profile of patients. However, a meta-analysis of Edwards and Moore (Edwards J.E. and Moore R.A.Statins in hypercholesterolaemia: A dose-specific meta-analysis of lipid changes in randomised, double blind trials. BMC Fam Pract. 2003; 4: 18) showed that different statins at a range of doses reduced total cholesterol by 17–35% and LDL-cholesterol by 24–49% from baseline. In our study in diabetic monkeys after 2 weeks of treatment, total cholesterol decreased from 9.25±1.11 mmol/l to 7.84±0.73 mmol/l (-15.2%), while LDL levels decreased from 6.64±1.31 mmol/l to 5.27±0.91 mmol/l (-21%). Interestingly HDL increased in both population after 4 weeks of treatment, but significantly only in non-diabetic one (in diabetic monkeys p=0.1 between baseline and 4weeks treatment). Taking into account the effect of treatment on lipid profile of both populations studied, the grade aspalathin-rich green rooibos extract showed a cardioprotective role highlighted by a LDL-lowering effect in diabetics and a HDL-increasing effect in both groups (significantly in non diabetic one).
Figure 2A, it seems there is a great decrease of blood glucose in the diabetic monkeys, without major changes in insulin production. As for glucagon, the authors say there was no difference at the baseline. I understand that the difference might not be statistically significant, but the glucagon levels in diabetic monkeys is 2X of that of the non-diabetic group. I would take a more cautious approach and indicate that there was a difference which didn’t reach statistical significance. I think these results warrant future studies investigating
Accordingly to the reviewer comment, the sentence has modified in lines 165-166